# Functional Characterisation of the Rare *SCN5A* p.E1225K Variant, Segregating in a Brugada Syndrome Familial Case, in Human Cardiomyocytes from Pluripotent Stem Cells

**DOI:** 10.3390/ijms24119548

**Published:** 2023-05-31

**Authors:** Nicolò Salvarani, Giovanni Peretto, Crasto Silvia, Andrea Villatore, Cecilia Thairi, Anna Santoni, Camilla Galli, Paola Carrera, Simone Sala, Sara Benedetti, Elisa Di Pasquale, Chiara Di Resta

**Affiliations:** 1Institute of Genetic and Biomedical Research (IRGB), UOS of Milan, National Research Council of Italy, 20138 Milan, Italy; nicolo.salvarani@humanitasresearch.it; 2IRCCS Humanitas Research Hospital, Via Manzoni 56, Rozzano, 20089 Milan, Italy; si.crasto@gmail.com (C.S.); cecilia.thairi@humanitasresearch.it (C.T.); camilla.galli@humanitasresearch.it (C.G.); 3Department of Cardiac Electrophysiology and Arrhythmology, IRCCS San Raffaele Scientific Institute, 20132 Milan, Italy; peretto.giovanni@hsr.it; 4Faculty of Medicine, Vita-Salute San Raffaele University, 20132 Milan, Italy; a.villatore@studenti.unisr.it (A.V.); diresta.chiara@hsr.it (C.D.R.); 5Department of Biomedical Sciences, Humanitas University, Via Rita Levi Montalcini 4, Pieve Emanuele, 20072 Milan, Italy; 6Genomic Unit for the Diagnosis of Human Pathologies, IRCCS San Raffaele Hospital, 20132 Milan, Italy; anna@annasantoni.it (A.S.); carrera.paola@hsr.it (P.C.); sarabe69@yahoo.it (S.B.); 7Laboratory of Clinical Molecular Biology, IRCCS San Raffaele Hospital, 20132 Milan, Italy

**Keywords:** Brugada syndrome, sodium current, cardiomyocyte

## Abstract

Brugada syndrome (BrS) is an inherited autosomal dominant cardiac channelopathy. Pathogenic rare mutations in the *SCN5A* gene, encoding the alpha-subunit of the voltage-dependent cardiac Na^+^ channel protein (Nav1.5), are identified in 20% of BrS patients, affecting the correct function of the channel. To date, even though hundreds of *SCN5A* variants have been associated with BrS, the underlying pathogenic mechanisms are still unclear in most cases. Therefore, the functional characterization of the *SCN5A* BrS rare variants still represents a major hurdle and is fundamental to confirming their pathogenic effect. Human cardiomyocytes (CMs) differentiated from pluripotent stem cells (PSCs) have been extensively demonstrated to be reliable platforms for investigating cardiac diseases, being able to recapitulate specific traits of disease, including arrhythmic events and conduction abnormalities. Based on this, in this study, we performed a functional analysis of the BrS familial rare variant NM_198056.2:c.3673G>A (NP_932173.1:p.Glu1225Lys), which has been never functionally characterized before in a cardiac-relevant context, as the human cardiomyocyte. Using a specific lentiviral vector encoding a GFP-tagged *SCN5A* gene carrying the specific c.3673G>A variant and CMs differentiated from control PSCs (PSC-CMs), we demonstrated an impairment of the mutated Nav1.5, thus suggesting the pathogenicity of the rare BrS detected variant. More broadly, our work supports the application of PSC-CMs for the assessment of the pathogenicity of gene variants, the identification of which is increasing exponentially due to the advances in next-generation sequencing methods and their massive use in genetic testing.

## 1. Introduction

Brugada syndrome (BrS) is an inherited autosomal dominant cardiac channelopathy, first described in 1992 and displaying incomplete penetrance [1]. BrS is characterized by cardiogenic syncope in otherwise healthy subjects and a typical electrocardiographic (ECG) pattern with ≥2 mm ST-segment elevation in the right precordial leads (V1–V3) and right bundle branch block, with an increased risk of malignant ventricular arrhythmias [2].

BrS has long been considered as an autosomal dominant disorder with incomplete penetrance; however, a more complex mode of inheritance has been recently proposed, which is associated with the presence of both rare and common variants in different genes [3,4]. 

BrS prevalence is estimated to be from 1/5000 to 1/2000, with a strong male predominance. In 20% of BrS patients, the clinical phenotype is associated with loss-of-function rare mutations in the *SCN5A* gene, encoding the α subunit of the voltage-dependent cardiac Na^+^ channel protein (Nav1.5) [5,6]. The clinical spectrum of *SCN5A*-positive BrS patients is characterized by a heterogeneous phenotype, although these patients generally exhibit more severe conduction abnormalities [7,8] and have worse arrhythmic outcomes compared to *SCN5A*-negative patients [3,4,9]. However, the presence of *SCN5A* pathogenic variants does not by itself justify prophyslactic ICD implantation, which currently represents the only available treatment. Nevertheless, given the risk of conduction disturbances, the occurrence and type of *SCN5A* pathogenic variants should be considered in patient risk stratification, in addition to the prognostic factors outlined in the presence of clinical risk markers, such as arrhythmogenic syncope [3,4,10]. 

Recently, the American College of Medical Genetics and Genomics (ACMG) recommended that incidentally detected pathogenic (class V) or likely pathogenic (class IV) *SCN5A* variants can be considered actionable and reportable [11]. However, in most cases, BrS patients carry private variants, whose pathogenic role is mainly assessed by exploiting time-consuming in vitro functional analysis, which is crucial for a better classification of newly identified rare *SCN5A* variants [12]. 

The assessment of the pathogenicity of new variants often needs in-depth characterization in order to ascertain a causal role in the disease. In this scenario, the application of human induced pluripotent stem cells has allowed an extraordinary improvement of our knowledge of human diseases, leading to the creation of numerous in vitro patient-specific cellular models for the study of new molecular mechanisms and new therapeutic strategies [13]. 

In the cardiac field of study, one possible approach is the exploitation of patient-specific isogenic cardiac models, which are generated by induced pluripotent stem cells (iPSC) and gene editing technologies. They would represent the most accurate platforms for testing the functional significance of a newly identified variant. However, patient cell reprogramming and gene editing approaches are extremely costly and time-consuming and thus would limit their broad application to assessing the pathogenicity of isolated genetic variations emerging from genetic tests.

On the other hand, human cardiomyocytes (CMs) differentiated from pluripotent stem cells (PSCs) have been extensively demonstrated to be reliable platforms for investigating cardiac diseases and being able to recapitulate specific traits of disease, including arrhythmic events and conduction abnormalities [13,14,15,16,17]. Moreover, several reports have previously demonstrated the possibility of either inducing or rescuing the main disease traits by viral-mediated (e.g., adeno-associated or lentiviral) overexpression strategies [14,15,18].

Based on these premises, in this study, we performed a functional analysis of the BrS familial rare variant NM_198056.2:c.3673G>A (NP_932173.1:p.Glu1225Lys) in human cardiomyocytes differentiated from pluripotent stem cells (PSCs). A schematic representation of the overall experimental work is provided in Figure 1. 

The putative damaging effect of this variant has been recently suggested by Glazer A.M. et al. using an automated high-throughput patch-clamp approach [19], but it has never been functionally characterized before at the level of the human cardiomyocyte. Therefore, in our study, by using a specific lentiviral vector encoding a GFP-tagged *SCN5A* gene carrying the specific c.3673G>A variant and CMs differentiated from control PSCs (PSC-CMs), we demonstrated a reduction of peak sodium currents in those cells specifically expressing the *SCN5A* variant, thus indicating a functional impairment of the corresponding mutated protein Nav1.5 and the pathogenicity of the rare BrS familial variant p.E1225K. Our results strengthen the proof of the pathogenic effect of this variant.

In addition, this study also supports the application of PSC-CMs as a versatile platform for the rapid assessment of the pathogenicity of variants that emerge from genetic analyses, a field of study that is expected to grow exponentially due to the recent advances in next-generation sequencing methods and their increased use for genetic testing [20,21].

## 2. Case Description

### 2.1. Clinical Evaluation

The index case and his familial members (parents and brother) underwent standard cardiological examination, including standard 12-lead ECG, 24 h Holter monitoring, and transthoracic echocardiogram. The proband and his brother underwent flecainide infusion (2 mg/kg) over 10 min, with continuous ECG monitoring. In the index case, invasive electrophysiological testing was performed at a referral center for arrhythmia management, through a standard protocol of programmed ventricular stimulation (PVS) up to three extrastimuli by right ventricular apical pacing [22]. Endoepicardial electroanatomical mapping of the right ventricular outflow tract (RVOT) was performed by multielectrode mapping catheters and a 3D navigation system (CARTO) to identify abnormal electrograms consistent with Brugada syndrome [23]. Catheter ablation was performed by radiofrequency energy delivery aimed to achieve the endpoint of full late potential abolition and noninducibility of sustained ventricular arrhythmias [24]. Regular follow-up was performed at specialized outpatient clinics with multidisciplinary facilities for Brugada syndrome. The follow-up length for the proband was 16 years. 

Written informed consent for genetic analysis, approved by the San Raffaele Institutional Health Department, was signed by all patients for diagnostic testing. The Research Protocol (ID: BS-01) was approved by the San Raffaele Institutional Ethical Review Board. The investigation conformed with the principles outlined in the Declaration of Helsinki.

### 2.2. Genetic Investigation

Written informed consent for genetic analysis approved by our Institutional Health Department was signed by all patients. The investigation conformed with the principles outlined in the Declaration of Helsinki. Genomic DNA was extracted from peripheral blood and the *SCN5A* coding sequence was analyzed by direct sequencing. Genomic DNA was extracted with the Maxwell^®^16 System (Promega_Italy, Milano, Italy) and all *SCN5A*-coding exons were amplified with primers in the intronic flanking region by polymerase chain reaction (PCR) [25]. Mutation analysis was performed using direct automated DNA sequencing with the ABI 3730 automatic DNA Sequencer (Applied Biosystems_Thermofisher, Waltham, MA, USA). The sequencing data were analyzed using the Sequencher software v5 (Gene Codes, Ann Arbor, MI, USA). Variants were annotated according to the GnomAD database [26], dbSNP v152 [27], and in-house sequencing database. For the prioritization and interpretation of the detected variant, we exploited in silico prediction tools, such as CADD [28], PolyPhen [29], SIFT [30], and Mutation Taster [31]. The conservation score was calculated by GERP11 [32]. The variant classification was performed according to the ACMG criteria [21]. 

### 2.3. Generation of pLenti-EF1alpha-SCN5A-GFP p.E1225D Lentiviral Vector

The lentiviral vector encoding the mutant *SCN5A* gene was generated by site-direct mutagenesis of the pLenti-EF1alpha-SCN5A-GFP_WT (original vector from OriGene—CW104036—a lentiviral construct encoding the C-terminus GFP-tagged wild-type SCN5A), using PCR-based site-directed mutagenesis (QuickChange XL Site-Directed Mutagenesis kit [Agilent Technologies, Santa Clara, CA, USA]) with the following primers: 5′-GCAGTGGAGCGCTGGCCTTCAAGGACATCTACCTAGAGGAG-3′ (forward) and 5′-CTCCTCTAGGTAGATGTCCTTGAAGGCCAGCGCTCCACTGC-3′ (reverse). All constructs were sequenced to verify the presence of the mutations and rule out spurious substitutions. 

### 2.4. Lentiviral Particles Production and Cardiomyocytes Transduction

Lentiviral particles encoding either wild-type or E1225K *SCN5A*-GFP genes were produced into HEK293T cells by calcium phosphate transfection, in the presence of the packaging pPax2 and the envelope VSV-G constructs, as previously described [33]. Lentiviral particles were collected at 30 h in the cardiomyocyte culture medium, and then used for cell transduction. For the overexpression experiments, d18–d20 human PSC-CMs were exposed to lentiviral particles in the presence of polybrene (10 µg/mL). Two cycles of infection (6 h and overnight) were carried out; the day after, cells were plated onto laminin (5 μg/mL)- and fibronectin (5 μg/mL)-coated glass coverslips for the electrophysiological measurements, as described in Section 2.6 below.

### 2.5. Human Pluripotent Stem Cells Maintenance and Differentiation

Experiments were performed by using cardiomyocytes (CMs) differentiated from the pluripotent stem cell (PSC) line RUES2 (hPSCReg Name: RUESe002-A), kindly provided by Prof. Ali Brinvanlou (Laboratory of Molecular Vertebrate Embryology, The Rockefeller University, New York, NY, USA). PSCs were maintained under 5% CO_2_ in StemFlex medium on Vitronectin (both from ThermoFisher Scientific, Waltham, MA, USA), as previously described [15]. Passaging was performed using 0.5 mM EDTA after a pre-treatment with 10 μM Rho-associated protein kinase inhibitor (Y-27632 from Selleck Chemicals, Planegg, Germany) when a confluency of about 85–90% was reached (~every 4 to 5 days). Differentiation into CMs was achieved using a chemically-defined serum-free monolayer protocol, which was based on the modulation of the Wnt pathway, as previously reported [34]. In brief, PSCs were dissociated with Accutase (Gibco-ThermoFisher Scientific, Waltham, MA, USA) and plated at a density of 0.25–0.5 × 10^5^ cm^2^ in 12-well plates coated with growth factor reduced (GFR) Matrigel (BD Biosciences, Franklin Lakes, NJ, USA) in StemFlex medium. Cells were grown in a 5% CO_2_/air environment until they reached 95–100% confluence (~4 days) and then induced for 24 h with the GSK3b-inhibitor CHIR99021 (12 μM, from Selleck Chemicals), which mediates Wnt activation (the day of the CHIR99021 treatment was referred to as day 0). At day 3, cells were then exposed to the Wnt inhibitor IWR-1 (5 μM from Sigma Aldrich, St. Louis, MO, USA) for 48 h, and media was replaced on day 5 and day 7. Treatments and media replacements were performed in RPMI supplemented with B27 without insulin. At day 10, the medium was supplemented with insulin. Spontaneous contracting activity usually appears from days 7 to 10 of induction and CMs are used for experiments 25 to 30 days after spontaneous contraction has started, after verifying a differentiation efficiency greater than 85% by FACS analysis using the cardiac-specific marker α-sarcomeric actinin. If lower differentiation efficiencies were obtained, a purification step was added by depleting non-cardiomyocyte cells using the PSC-derived Cardiomyocyte Isolation Kit (human, cat# 110-130-188 from Miltenyi Biotec, Bergisch Gladbach, Germany).

### 2.6. Functional Characterization: Assessment of I_Na_ Currents by Patch-Clamp 

Whole-cell Na^+^ current (*I*_Na_) recordings were performed in low-density cell cultures, grown on glass coverslips (VWR International, Radnor, PA, USA). Cultures were transferred to a custom-made experimental chamber 2 to 3 days after seeding and recordings were performed at 36° Celsius using a Multiclamp 700B patch-clamp amplifier (Molecular Devices, San Jose, CA, USA) controlled by pClamp 10.3 software (Molecular Devices). For the analyses, dedicated software (ClampFit 9.2) was used. *I*_Na_ was perfused at 1.5 mL min^−1^, low-pass filtered at 10 kHz, digitized at 20 kHz, and stored for offline analysis, whereupon the latter was performed using dedicated software (pClamp 10.6 and Clampfit 10.6; Molecular Devices). Patch pipettes were pulled from borosilicate glass capillaries (Intrafil-10, INTRACEL LTD – Abbotsbury Engineering Ltd, St. Ives, United Kingdom) with a laser-based micropipette puller (P-2000; Sutter Instrument, Novato, CA, USA) and resistances ranging from 1 to 3 MΩ. After seal formation (2–10 GΩ), rupturing of the patch, and before *I*_Na_ measurement, adequate voltage control was achieved with membrane capacitance (C_m_) and series resistance (R_s_: 3–8 MΩ) compensation above 80% (estimated voltage error <5 mV in all cases), and by using a holding potential of −90 mV for 300 ms. Whole-cell *I*_Na_ recordings were conducted in an extracellular solution containing (in mmoL^−1^): NaCl 70, CaCl_2_ 1.8, CsCl 90, MgCl_2_ 1.2, glucose 10, HEPES 10, and Nifedipine 0.001 (pH 7.4 with CsOH) and pipette solution filled with (in mmoL^−1^): CsCl 130, NaCl 10, CaCl_2_ 1, EGTA 10, HEPES 5, Mg-ATP 3 (pH 7.2 with CsOH). Nifedipine was added to the external solution to block I_CaL_, while K^+^ was substituted for Cs^+^ in both the pipette and external solution to prevent current flow through K^+^ channels [15].

Potentials were corrected for the estimated change in liquid junction potential of 6 mV before the experiments. Peak *I*_Na_ densities and the voltage-dependence of activation were characterized as follows: single cells were held at −90 mV and 100 ms voltage steps were applied from −90 to +80 mV in 5 mV increments with a cycle time between voltage steps of 3 s. Voltage-dependence of inactivation was assessed by holding cells at various potentials from −130 to 15 mV for 150 ms followed by a 50 ms test pulse to −20 mV to elicit *I*_Na_ (holding potential of −90 mV). To consider the variations in cell size, the current amplitude was normalized to C_m_ and expressed as current density (pA/pF). Activation and inactivation curves were described by fitting experimental points with the Boltzmann equation to estimate the voltages of half maximal activation and inactivation (V_1/2_) and each slope factor (k). Statistical comparisons were made using the Student’s t-test for unpaired samples. Differences with at least *p* < 0.05 were regarded as significant.

### 2.7. Clinical Case

#### 2.7.1. Clinical Features

A 24-year-old male (II.2) (Figure 2A) was referred to the Arrhythmology Unit of the IRCCS San Raffaele Hospital for an unexplained and sudden syncope. He received a clinical diagnosis of BrS, based on the presence of the type-1 spontaneous BrS pattern at baseline ECG. The proband had no known familial history of sudden cardiac death (SCD) and an unremarkable past medical history. The baseline echocardiogram was unremarkable. The proband’s programmed ventricular stimulation (PVS) resulted positive for polymorphic ventricular tachycardia (VT) from the right ventricular outflow tract (RVOT), degenerating into ventricular fibrillation (VF) terminated by direct current (DC) shock. Remarkably, atrial tachycardia (AT) with variable conduction up to 1:1 was also induced. Subsequently, he was admitted to a third-level center for arrhythmology. A new PVS, with up to three extra stimuli until the refractory period, resulted negative. Since the type-1 BrS pattern was not spontaneously present at that time, a pharmacological test was performed. After flecainide infusion, the BrS pattern was enhanced, and recurrent polymorphic VT episodes emerged due to drug toxicity (Appendix A). A transvenous implantable-cardioverter defibrillator (ICD) was implanted in secondary prevention. During follow-up, he was readmitted for multiple appropriate ICD shocks on pre-syncopal VT, which frequently occurred after heavy alcohol consumption and emotional stress. He underwent endocardial–epicardial electroanatomical mapping, with epicardial substrate radiofrequency ablation. Two years later, however, VT recurred and hydroquinidine was started, only with short-term benefit on the arrhythmic burden. A second ablation was tempted, but new VT events occurred after the procedure. Finally, hydroquinidine was resumed, achieving long-term arrhythmic control. Due to positive genetic screening, the 28-year-old brother (II.1) (Figure 2A) underwent clinical investigation; despite an unremarkable baseline ECG, he tested positive at flecainide challenge and received an ICD in primary prevention. At the follow-up, he remained asymptomatic for syncope, and no ventricular arrhythmias were ever detected at ICD interrogation. Their 56-year-old father (I.1) (Figure 2A), despite showing a type-2 Brugada pattern and carrying the same *SCN5A* variant, refused to undergo further investigation and had an uneventful follow-up.

#### 2.7.2. Genetic Testing Result 

The proband carried the heterozygous missense variant NM_198056.2 (*SCN5A*):c.3673G>A (NP:932173.1:p.Glu1225Lys), inherited from the father and present also in his brother (Figure 2B). The detected variant was previously reported as rs199473204 in NCBI-dbSNP and described in ClinVar in some BrS and long QT syndrome (LQTS) cases (VCV000067810.9) [35]. Its frequency in the GnomAD database was <1:100.000, its deleterious effect was predicted by most in silico tools, and the Glu residue was highly conserved among species (Figure 2C). The amino acid alteration was localized in the extracellular loop between segments 1 and 2 of the III repeat domain of the protein (Figure 2D). The variant was thus classified as likely pathogenic (class IV), according to the following ACMG criteria: PS3 (in vitro functional data from the heterologous system) [19], PM2 (absent in controls), PP3 (multiple lines of computational evidence support a deleterious effect), PP5 (reputable source recently reports variant as pathogenic).

#### 2.7.3. Functional Testing Indicates Reduced Peak Sodium Currents in CMs Overexpressing p.E1225K Na_v_1.5 Channel

In order to unequivocally determine whether the rare *SCN5A* variant identified in the BrS family described above can impact the activity of the respective sodium channel Nav1.5, and thus was causative of the disease, we overexpressed the p.E1225K-*SCN5A* in CMs obtained from the human RUES-2 PSC line (PSC-CMs) and analyzed them at the functional level, through the assessment of Nav1.5 currents by patch-clamp. Of note, PSC-CMs currently represent the most physiologically relevant human cardiac cellular model for in vitro studies. Specifically, we generated a C-terminus GFP-tagged lentiviral construct encoding the *SCN5A* gene carrying the p.E1225K BrS variant (Figure 3A); as a control, the same vector but expressing the wild-type form of the gene (pLenti-EF1alpha-SCN5A-GFP_WT, from Origene) was used. For the overexpression experiments, lentiviral particles expressing either WT or p.E1225K *SCN5A* were produced (Appendix A) and then used to transduce the PSC-CMs; measurements of sodium currents were then performed on the GFP-positive cells that were specifically overexpressing the WT (WT-GFP) or mutant (MUT-GFP) *SCN5A*. The results from the patch-clamp recordings showed a significant decrease in average peak *I*_Na_ density in the MUT-GFP CMs with respect to the control WT-GFP condition (Figure 3B), with a reduction of 67.1% at a membrane potential of −30 mV (Figure 2D). Voltage dependences of activation and inactivation parameters (half maximal voltage [V1/2] and slope factor [k]) measured between the two conditions were not significantly different (*p*-value: 0.077 and 0.054), albeit the average *I*_Na_ currents recorded in WT-GFP and MUT-GFP CMs peaked at different voltages, respectively, at −25 mV and −20 mV (Figure 3B). For clarity, a positive shift of the MUT-GFP activation curve with respect to the WT-GFP was detected; however, this was found to be significant only at −30 and −25 mV (Figure 3C). These results may potentially depend on technical and/or biological reasons. First of all, a technical aspect concerning patch-clamp experiments on fast voltage-gated sodium channels should be considered [36]; series resistance (Rs) values were higher in WT-GFP than in MUT-GFP CMs (5.56 ± 1.72 MΩ vs. 4.69 ± 1.20 MΩ n.s.) and this generated a worse control of the command potential due to a voltage drop across Rs. Secondly, both WT-GFP and MUT-GFP CMs expressed not only the transduced protein but also the endogenous Nav1.5 currents, which may have influenced the current generated by the transduced mutant channel, thus explaining the shift in the activation curve. However, we cannot exclude that this effect could have been specifically mediated by our variant, as has been reported for other missense mutations associated with BrS [37]. 

Of note, Nav1.5 currents and voltage dependencies measured in the WT CMs were in line with published values measured in human CMs obtained from PSCs differentiation, either in our or other laboratories [38], further proving the validity of our experimental setting and results. 

Altogether, the results described in this work showed a functional impairment of the sodium channel protein Nav1.5 carrying the p.E1225K variant, thus supporting its functional association with the pathogenesis of BrS. 

## 3. Discussion

In this study, we provide the first in vitro functional characterization in human CMs of the rare aminoacidic substitution p.E1225K in *SCN5A,* identified in a BrS case, with a silent family history for BrS. The diagnostic *SCN5A* genetic testing was fundamental for the risk stratification of the proband and his family. Indeed, to date, it is recognized that *SCN5A* mutations are associated with the risk of lethal arrhythmia in BrS [4]. Moreover, the subsequent study segregation allowed the identification of two familial members (I.1 and II.1) carrying the same genetic alteration, who were potentially at risk of developing the malignant BrS phenotype and needed further clinical evaluation. 

To date, even though hundreds of *SCN5A* variants have been associated with BrS, the underlying pathogenic mechanisms are still unclarified in most cases [39]. Therefore, the functional characterization of the *SCN5A* BrS rare variants still represents a major hurdle; thus, it is fundamental to confirm their pathogenic loss-of-function effect, which is also an important requirement for the assignment of the ACMG criteria [21]. Indeed, nearly 70% of BrS *SCN5A* rare variants registered in ClinVar are reported as variants of unknown significance (class III), requiring further functional evidence to classify them as causative BrS mutations, according to the ACMG criteria. Consequently, since the identification of a pathogenic mutation could be useful for predicting patient prognosis and guiding therapeutic choices [40,41], an important challenge is to accurately classify variants, especially when only limited phenotype data are available [42]. Indeed, according to ACMG guidelines, the evidence score of pathogenicity for “absent in population databases” is assigned as moderate (PM2), while that of “in silico-prediction supporting” is assigned as PP3. However, it is mandatory to consider that the specificity of in silico algorithms for predicting the pathogenicity of missense variants is quite low despite their high sensitivity [21], resulting in the overprediction of missense variations as deleterious. Recent studies based on purely in silico analyses have failed to predict the disease risk of *SCN5A* variants in BrS, using the ACMG guidelines [43,44]. These results support the implications of additional reliable functional evaluations of *SCN5A* variants, exploiting patch-clamp studies, to improve the discrimination between causative and benign rare *SCN5A* variants [40]. Indeed, “functional studies supporting a deleterious effect (PS3)” constitute one of four criteria with “strong evidence” of pathogenicity [21]. 

Therefore, in this study, we carried out the first in vitro electrophysiological characterization of the rare p.E1225K *SCN5A* variant, applying a standard voltage-clamp approach to CMs derived from human pluripotent stem cells using the patch-clamp technique. The main effect of the p.E1225K mutation that we detected in this study was a substantial decrease in *I*_Na_ density compared to the wild-type condition. Thus, the association between the p.E1225K mutation and the BrS phenotype of the familial case described here may be primarily due to a reduction of *I*_Na_ density, which leads us to postulate conduction abnormality as the most likely mechanistic model. Indeed, it is well known that Nav1.5 voltage-gated sodium channels are mainly responsible for the upstroke phase of the action potential and key regulators of the electrical impulse propagation across the myocardium [45]. Therefore, while the underlying BrS molecular mechanisms still need to be completely determined, the reduction of peak *I*_Na_ currents is expected to lead to reduced upstroke velocity in CMs carrying the mutation, and coherently to also be associated with conduction disturbances, which typically occur in BrS patients. It is also important to report that recent studies further support a model in which total *I*_Na_ peak current is correlated with disease severity [41]. It is worth mentioning that in some BrS patients, repolarization abnormalities might be present [39].

Recently, Glazer et al. exploited the automated patch-clamping approach for a high-throughput study of several rare *SCN5A* variants [19], including the p.E1225K. Our data are in line with those obtained from Glazer and colleagues, which show a peak *I*_Na_ current density reduced to 32.9% compared with the control condition; importantly, the effect registered on sodium current for our p.E1225K variant was significantly more dramatic than those reported for the most common mutation associated with BrS, the p.E1784K, which further supports the relevance of our variant in the disease context [46]. However, these studies were based on a heterologous expression system, namely overexpression in HEK293T, which are immortalized human embryonic kidney cells and, as such, devoid of most of the cardiac-specific proteins expressed in human cardiomyocytes, potentially interacting with Nav1.5 and/or modulating its function. As a result, the potential pathogenic effect of some variants might be missed if analyzed in these experimental settings [47,48]. Therefore, albeit their known immature fetal-like phenotype [48], the use of PSC-CMs allows us to overcome this limitation and obtain information on the functional effect of a variant in a cardiac-specific context. Moreover, compared with heterologous cellular models, PSC-CMs are excitable cells characterized by an action potential (AP), which is at the basis of the electrical activity of cardiac cells during contraction and relaxation of the heart. Therefore, the use of such models allows us to study the effect of gene variants on the AP-specific properties (maximal upstroke velocity, amplitude, duration, etc.), and on the specific ionic currents that contribute to each AP phase. In the specific case of the *SCN5A* p.E1225K variant investigated in this study, we expected to measure an effect on the upstroke velocity (dV/dt_max_), a phase in which the Nav1.5 channels play a key role, and the related properties (overshoot and AP amplitude). Of note, dV/dt_max_ is at the basis of proper impulse propagation in the cardiac tissue and its alteration may lead to conduction disturbances and arrhythmic events.

For the sake of clarity, it is worth noting that the PSC-CM-based model proposed here cannot be applied to the study of all variants, such as autosomal recessive ones. Indeed, in this case, the native endogenous expression of the cardiac-specific proteins under study would not allow us to reproduce the homozygous state of the recessive variant in the cells and thus would mask its effect. On the other hand, this approach is very well suited to the study of autosomal dominant variants, which are expressed in heterozygosity. In the described case, the analyzed autosomal dominant and heterozygous mutation segregates in familial affected members; however, its incomplete penetrance and expressivity are highly variable, as can be expected from data presented in the literature [2,49,50]. Indeed, BrS is characterized by incomplete penetrance and its clinical manifestation is influenced by different factors, such as age and sex [51], as well as common genetic variants, including multiple noncoding haplotypes near the *SCN5A* gene [52,53].

Thus, we can hypothesize that the phenotypic heterogeneity in the family may be explained by recent theory, supported by increasing evidence suggesting that BrS is not a classical mendelian monogenic disease but rather an oligogenic disorder involving multiple rare and nonrare variants, as well as structural abnormalities and inflammation, which contribute to the underlying basis of the disease [3,54]. However, the contribution of polygenic factors in BrS warrants further investigation, which could help to define a new personalized risk stratification paradigm for sudden cardiac death. Further studies, to be pursued in the future on cardiomyocytes generated from patient-derived induced pluripotent stem cells, may clarify this and shed light on the potential disease modifiers and related underlying mechanisms.

## 4. Conclusions

In this study, we provide the first functional characterization in human CMs of the rare aminoacidic substitution p.E1225K in Nav 1.5 by the use of a standard voltage-clamp approach. The main detected effect was a substantial decrease in *I*_Na_ density in CMs overexpressing the variant compared to the control condition, which means we can postulate conduction abnormalities as the most likely mechanistic BrS model. Therefore, our data further strengthen the proof of the pathogenic effect of this variant and its correlation with the BrS phenotype. 

However, it is important to consider that the BrS phenotypic variability and the contribution of polygenic factors warrant further investigation, to ameliorate the personalized risk stratification of patients. Therefore, further studies on cardiomyocytes generated from patient-specific induced pluripotent stem cells should be highly considered in the future, to better clarify phenotypic variability and shed light on potential disease modifiers and the related underlying mechanisms.

## Figures and Tables

**Figure 1 ijms-24-09548-f001:**
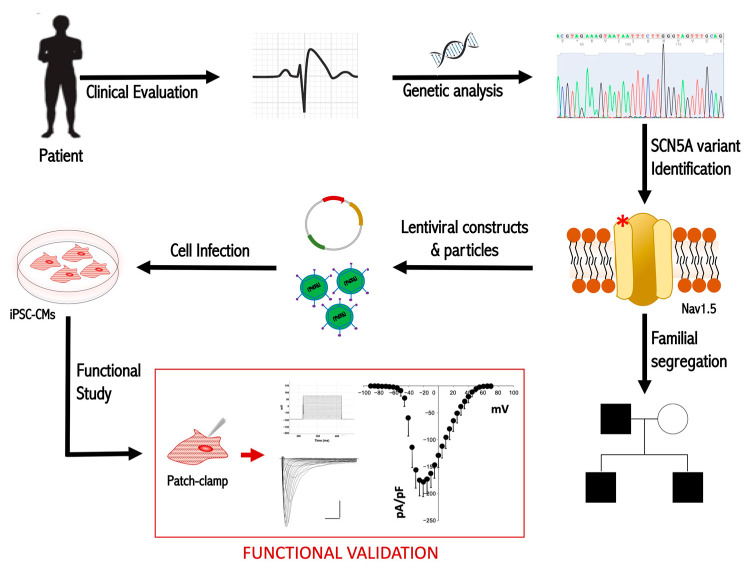
Graphical overview of the experimental work. The red asterisk (*) indicates the mutation on the Nav1.5 channel.

**Figure 2 ijms-24-09548-f002:**
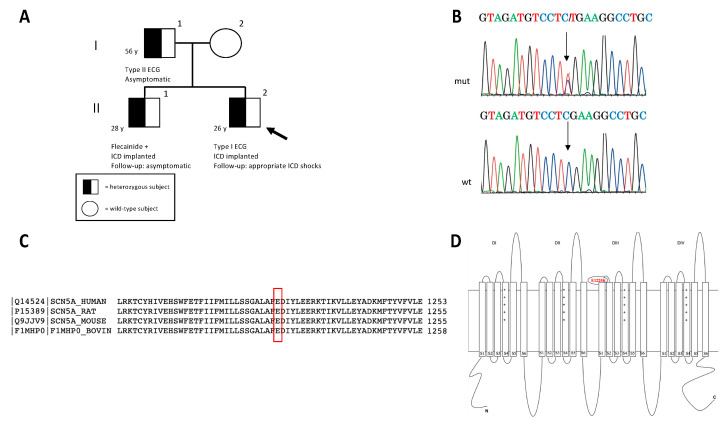
Pedigree structure and main features of the rare *SCN5A* detected variant. (**A**) In the pedigree, the arrow indicates the proband (proband’s age: two years later the first sudden syncope) and circles and squares indicate females and males, respectively. The genotype characterization of the family is depicted with an empty symbol (wild-type mother) and black/white symbol (mutated members). (**B**) The electropherogram of the mutated (mut) *SCN5A* gene sequence in the proband is reported and compared to the wild-type (wt) sequence; arrows indicate the nucleotide substitution c.3673G>A leading to the p.E1225K mutation. Reverse strands are reported. (**C**) The evolutionary conservation of Nav1.5-E1225 residue among different species (UniProtKB/Swiss-Prot). (**D**) Schematic representation of Nav 1.5 protein. The p.E1225K mutation was localized in the extracellular loop between segments 1 and 2 of the III repeat of the protein.

**Figure 3 ijms-24-09548-f003:**
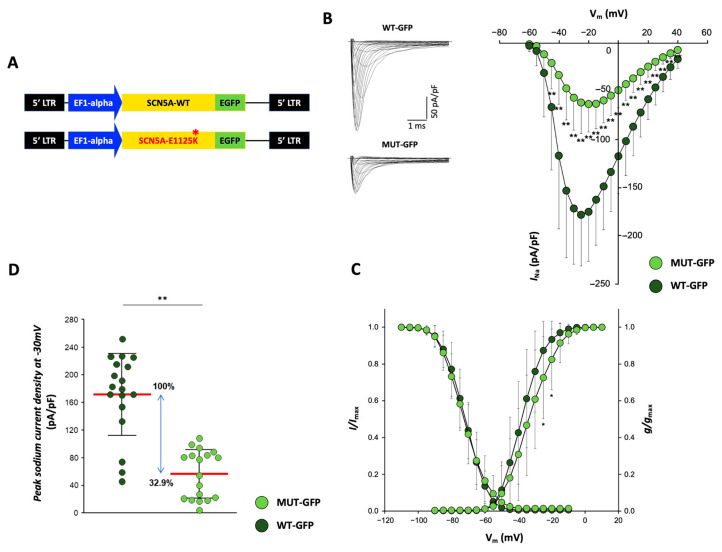
Overexpression of p.E1225K-Nav1.5 leads to reduced peak sodium currents in PSC-CMs. (**A**) Lentiviral vectors expressing the C-terminal GFP-tagged SCN5A (wild-type and carrying the p.E1225K variant) used to produce the lentiviral particles for overexpression. (**B**–**D**) Voltage-gated sodium currents in PSC-CMs. (**B**) Left panel: examples of Na^+^ current (*I*_Na_) traces recorded in CMs overexpressing SCN5A-WT (top—WT GFP) and SCN5A-E1225K (bottom—MUT GFP) (Scale bar, 1 ms, 50 pA/pF). Right panel: I-V curves constructed from average peak sodium current density as a function of voltage command measured in WT GFP-CMs and MUT GFP-CMs, showing a significant reduction in the latter. WT GFP-CMs: n = 18; MUT GFP-CMs: n = 18. (**C**) Voltage dependences: steady-state activation (WT GFP-CMs: n = 18; MUT GFP-CMs: n = 18)/inactivation curves (WT GFP-CMs: n= 18; MUT GFP-CMs: n = 16). All values are reported as mean ± SD. * *p* < 0.05, ** *p* < 0.005 (unpaired *t*-test). Data are relative to four independent cell differentiations. (**D**) Dot plot of peak sodium current densities, measured at −30 mV, in MUT GFP-CMs (56.53 ± 33.96 pA/pF) relative to WT GFP-CMs (171.73 ± 57.83 pA/pF). Average current density values are indicated in bold red. The blue arrow indicates the average *I_Na_* density values, expressed as a percentage.

## Data Availability

https://doi.org/10.17632/p2zt42p4f6.1 (accessed on 18 April 2023).

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
