# Peer review of "Functional Characterisation of the Rare SCN5A p.E1225K Variant, Segregating in a Brugada Syndrome Familial Case, in Human Cardiomyocytes from Pluripotent Stem Cells"

_ijms, 2023, doi:10.3390/ijms24119548_

Round 1

Reviewer 1 Report

This study discovered a rare Brugada syndrome-associated variant in patients and then characterized the function of this variant in human induced pluripotent stem cells. They found that this variant leads to reduced peak sodium current suggesting the pathogenicity of this BrS variant.

However, the structure of this manuscript is kind of confusing. It looks like a combination of a case report and experiments. This manuscript is allegedly a case report, but the whole study weighed the experiments relative to the case description and treatment. So I think this manuscript slightly moved the focus from the case to the experiment. Although sometimes the case report could be prospective and experimental, the focus should be still the case. Besides, according to the guidelines of case report in IJMS, no experiments should be involved in a case report.

Author Response

We perfectly got the point raised by the Reviewer and we thank you for the suggestion of reconsidering the structure of this manuscript. We submitted it as a clinical case, as suggested by the Editorial Office, since the experimental data are strictly aimed at supporting the causal relationship between the detected rare variant and the reported clinical phenotype. However, we could further evaluate this point with them, proposing the manuscript as Clinical Case or Original Article.

Reviewer 2 Report

This manuscript is interesting and can be accepted after minor changes as indicated here: Improve the introduction with more background about the topic  Add the novelty of this work  Add a schematic diagram to show the overall work  Add some supporting data apart from the given test to confirm the results  The quality of the figures could be better  Add conclusion and future prospects of the work Also carefully rectify linguistic and typos errors. 

Check the grammar and typos.

Author Response

We thank the Reviewer for the suggestions. We welcome the good suggestions, improving all the raised points.  

  1. We added more details in the Introduction.
  2. We increased the visibility of the novelty of our work, as reported at the end of the introduction.
  3. We added a schematic diagram and a short paragraph on study design to better clarity it. Thank you.
  4. The causal role of the detected variant is supported by previous published works in heterologous systems (cited in the text) and by the well-established role of SCN5A as main causative gene in BrS.
  5. We tried to improve the figures, where it was possible.
  6. We added a short paragraph for the conclusions and future perspectives.
  7. We corrected the text.

Reviewer 3 Report

Dear authors, I am happy to have been given the opportunity to review this paper. I find the article to be very interesting, well structured and well documented. I only have minor suggestions, listed below:

1. Move the Discussion chapter after the Material and method chapter.

2. I believe the authors have forgotten to remove some text belonging to the IJMS document that describes how to structure your paper: for instance, lines 192-194, 425-439

3. Please clarify for the readers: is the SCN5A p.E1225K variant a VUS, or not? Please elaborate on this topic.

4. I am not sure references 43 and 44 are properly cited.

Author Response

We thank the Reviewer for the suggestions. We welcome the good suggestions, improving all the raised points. 

    1,2, 4. We modified it.

  1. We modified, as suggested, adding more information about the ACMG variant classification as likely pathogenic (class IV).

Reviewer 4 Report

Dear Authors,

Congratulations to the manuscript. The topic is important, up to date, the article is excellent.

Please check carefully and correct (until Line 439 and supplementary mentioned here) these below:

Abstract

Line 26                  associated with

Line 26      sg similar might be a bit better:

„Pathogenic rare variants in SCN5A (mutations) are identified in 20% of BrS families..

(from :  Wijeyeratne YD, Tanck MW, Mizusawa Y, ………….Behr ER. SCN5A Mutation Type and a Genetic Risk Score Associate Variably With Brugada Syndrome Phenotype in SCN5A Families. Circ Genom Precis Med. 2020 Dec;13(6):e002911. doi: 10.1161/CIRCGEN.120.002911. Epub 2020 Nov 9. PMID: 33164571; PMCID: PMC7748043.)

Line 28     „ even though thousands „   please correct (if needed) to:   „even though hundreds”

(Please check on page 673. :

Antzelevitch C, Yan GX, Ackerman MJ, Borggrefe M, Corrado D, Guo J, Gussak I, Hasdemir C, Horie M, Huikuri H, Ma C, Morita H, Nam GB, Sacher F, Shimizu W, Viskin S, Wilde AAM. J-Wave syndromes expert consensus conference report: Emerging concepts and gaps in knowledge. Europace. 2017 Apr 1;19(4):665-694. doi: 10.1093/europace/euw235. PMID: 28431071; PMCID: PMC5834028.)

Introduction

Line 52:      considered as

Line 54:      associated with

Line 72:     please check space between the words:   identified  rare  (seems much space)

Line 77:    thus would limit their

Line 77:  to assess pathogenicity

Figure 1     Panel A:  

In the text:

male II.2   is 24 years       , -in the same line the Panel A is indicated on which the patient is  26 (26 is when the genetic screening took place)

It turns out some rows later for the reader that the 26 is not mistyping on Panel A but was appropriately shown.

            Maybe you can solve in Line 99 by sg like this:  (Figure 1, Panel A: 2 years later)

Line 134 Figure 1 legend:   reverse strands are

Line 104:  please write out/explain PVS          programmed ventricular stimulation

Line 104   ventricular tachycardia (VT),  right ventricular outflow tract (RVOT)

Line 118   hydroquinidine

Figure 2              Panel B

INa curves                      You can consider putting the asterisks on MUT and not on WT as reduction is the effect

If you perform, the same is applicable to Panel C asterisks

why SD and why not    SEM is shown

please correct as the same you have written for *p<0.005, **p<0.005

Panel D      y values (ordinate) seem to be not correct   also unreadable, half is missing above 100

Line 139   presents

Line 152    point is not needed after channel

Line  157  PSC

Line 158   cardiac

Lines 168, 173, 175     Figure   (not Fig.)

Line 192     delete                         

Line 193     delete

Line 194      delete

Line 221     „ thousands”          please check: hundreds

Line 258:  You can insert 1 sentence sg like this or similar (optional)   :

    „ It is worth mentioning that in some BrS patients repolarization abnormalities might be present.”

       (from article /on its page 675, repolarization defect is also detailed/ below)  :

Antzelevitch C, Yan GX, Ackerman MJ, Borggrefe M, Corrado D, Guo J, Gussak I, Hasdemir C, Horie M, Huikuri H, Ma C, Morita H, Nam GB, Sacher F, Shimizu W, Viskin S, Wilde AAM. J-Wave syndromes expert consensus conference report: Emerging concepts and gaps in knowledge. Europace. 2017 Apr 1;19(4):665-694. doi: 10.1093/europace/euw235. PMID: 28431071; PMCID: PMC5834028.)

Line 260       in line with

Line 266     a heterologous

Line 269     As a result

Line 290     dominant

Line 293      incomplete penetrance

Line 306     delete                         

Line 307     delete

Line 308     delete

Line 309     delete

Line 312   Index case  = The patient and his ….             (you can choose)

4. Materials and Methods

Is this section is at proper order (4.), why not earlier?

Line 325     16 months?

Line 360 culture medium

Line 391.  Point is needed at the end of the sentence.

Line  393   cultures

Line 398  please check     INa were superfused (?)

Lines 425- 439  delete

Supplementary

Figure 1   panel B    please check   visibility of    lead names  e.g.  aVR  (I can see on magnification, maybe readers also will see upon magnification)

Figure 2:

      figure legend:   please insert space between 200mM  like: 200 mM

please check my recommendations I have written at the correction points (I indicated in the necessary places)

Author Response

We are grateful to the Reviewer for the kind comments. We modified the manuscript, according to the above suggestions. Changes are highlighted by the track changes.

Of note, regarding the specific comment on the follow-up, we can confirm that 16 years is correct.

We also checked visibility of Supplementary Figure 1B: we understand the reviewer’s point but we reported it at the best of our possibilities. We will ask technical support to the Editorial Office in case of publication.

Lastly, in the figure 2 we chose to use standard deviation (SD) instead of standard error of the mean (SEM) because in our opinion it is more appropriate to the type of data we are showing. SD measures data dispersion within a group, while SEM quantifies the uncertainty of the sample mean estimate compared to the population mean. SD is employed to analyze data variability and distribution, while SEM is commonly used for statistical inferences regarding the population mean or assessing the significance of differences between group means.

In summary, SD is a measure of dispersion that indicates the average deviation of data points from the mean in a dataset. It is used to assess variability within a dataset or measurements, provides insights into data dispersion, identifies phenotypic or genetic variability, and detects outliers. SD is valuable for analyzing data distribution, measuring variability within groups of individuals/measurements, and comparing variability between different experimental conditions.

Round 2

Reviewer 1 Report

Great work!